# CROSS-MODALITY IMAGE INTERPRETATION VIA CONCEPT DECOMPOSITION VECTOR OF VISUAL-LANGUAGE MODELS

## ABSTRACT

Inherently interpretable image classification is valuable for high-risk decision-making. Recent works achieve competitive performance against black-box models by combining visual language models (VLM) with concept bottleneck models (CBMs). Their explanations are achieved by the weighted sum of similarities between the image representation and embeddings of pre-defined texts. However, using text only is not sufficient to represent visual information and the choices of texts are subjective, resulting in potential compromises in both interpretations and performance. Therefore, this work explores cross-modality interpretation of critical concepts in image classification. Specifically, we build CBM with a set of decomposed visual concepts learned from images rather than pre-defined text concepts, namely decomposed concept bottleneck model (DCBM). The decomposition is implemented by vector projection to concept decomposition vectors (CDVs). To explain CDVs in different modalities, a quintuple notion of concepts and a concept-sample distribution are proposed. Experiments indicate a competitive performance of DCBM with non-interpretable models and superior interpretability compared to other CBMs in terms of sparsity, groundability, factuality, fidelity, and meaningfulness.

## 1 INTRODUCTION

Inherently interpretable models are attractive to high-risk decision-making (Rudin, 2019). Koh et al. (2020) introduced the concept bottleneck models (CBM), which breaks down the classification process into two parts: (1) predicting whether a sample contains a specific concept via multi-label classification, called concept scores. (2) predicting the final classification based on the concept scores, see Figure 1. Nevertheless, CBMs exhibit significant performance gaps compared to end-to-end neural networks and require dense concept annotations. To reduce the labeling effort, recent works combine CBM with the remarkable zero-shot classification performance of visual language models (VLMs) (Radford et al., 2021). They calculate concept scores as the similarities between image representations and embeddings of encoded pre-defined texts, making an expansion in concept numbers and increasing the classification performance (Yuksekgonul et al., 2022).

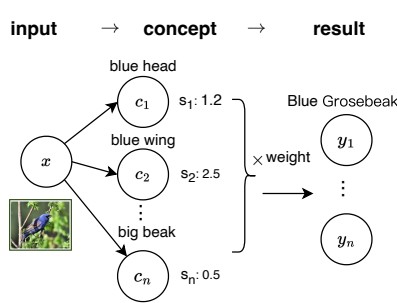

Figure 1: CBM workflow.

However, current VLM based CBMs might present two issues. Firstly, selecting an appropriate set of concepts is subjective, and manual construction may not cover all viusal features that are relevant to certain class. One possible solution is to use Large Language Model (LLM) to generate sufficient texts (Oikarinen & Nguyen, 2023; Yang et al., 2022), but this introduces redundant texts that are difficult to validate, thus compromises the reliability of interpretation. A failure case is shown in Figure 2, the concept set of "Blue Grosebeak" does not include "blue wing", while non-factual descriptions are present. Secondly, there exists a modality gap between image and text embeddings in VLMs (Liang et al., 2022), which leads to information loss in the zero-shot predicted concept score. As a result, it is also an important problem to explore how to prevent this information loss in both the classification and interpretation.

Figure 2: Schematic of concepts from different modalities in VLM latent space. Visual concepts are expected to be used for interpretation rather than text concepts given by LLM in this work.

In this paper, we build CBM using visual concepts from training images instead of pre-defined text concepts. As shown in Figure 2, the blue bird image is expected to be classified via decomposed visual concepts "blue and black head"($s_3$) and "blue wings"($s_4$) rather than text concepts associated with its label "Blue Grosbeak"($s_1$ and $s_2$). The decomposed visual concepts are represented by vectors called concept decomposition vectors (CDVs) and the proposed method is called as decomposed concept bottleneck models (DCBM). Taking the place of text embeddings using CDVs avoids the influence of modality gap, and the manual choices of texts would not influence the classification procedure. Technically, the CDVs are adversarially trained with a discriminator that distinguishes the CDVs from embeddings of training image. The adversarial training ensure CDVs To decrease information loss in interpretation, CDVs are interpreted in cross-modal manner, where the concepts can be visualized in images and expressed in natural language at the same time. To achieve cross-modality interpretation, concepts are formalized with quintuple notions, where each concept contains a CDV, a class name, a scalar weight, a set of image fragments, and a set of text. The samples that used to represent the CDV are selected from a categorical distribution, namely concept-sample distribution (CSD).

Experiments on various datasets indicate that DCBM achieves competitive performance against black-box classifiers. The quality of cross-modality interpretations are evaluated in three different levels to understand the model decision.how the visual and textual representations in VLMs are consistent with the same concept, which cannot be achieved by single-modal explanations alone. To compare DCBM with other VLM-based CBM, a human evaluation is conducted showing that DCBM explanations exhibit significant advantages. The contributions of this paper are summarized as follows:

- Propose the concept-sample distribution, which illustrates the goal of contrastive image-text matching and motivates a method of learning concepts across different feature spaces.
- Propose the concept decomposition vector (CDV) to decompose class-related visual concepts, and make DCBM less biased and more accurate.
- propose a novel form of cross-modality interpretation to explain decomposed concepts. From the concept level to the class level, it is able to reveal inherent biases and classification challenges for given datasets compared to the single modality explanation.

## 2 PRELIMINARIES

### 2.1 CONTRASTIVE IMAGE-TEXT MATCHING PRETRAINING OF VLMS

Visual Language Models (VLMs) are a series of models that can understand and generate both images and text, e.g. CLIP(Radford et al., 2021). Image text matching (ITM) is the common training objective that maps image representation into a language concept embedding space. We formalize the ITM training objective in the next paragraph.

Let $x \in \mathcal{X}$ denote an image $x$ in an image set $\mathcal{X}$, and $t \in \mathcal{T}$ represents a text $t$ in a text set $\mathcal{T}$. $\{(x_i, t_i)|i = 1, \ldots, N\}$ denotes $N$ image-text pairs, the match relationship can be represented with an identical matrix $\mathbf{Y}$, where $\mathbf{Y}_{ij} = \begin{cases} 1, & \text{if } i = j \\ 0, & \text{otherwise} \end{cases}$. In general, a VLM consists of an image encoder $I(\cdot)$, which maps the input image $x$ into a $d$-dimensional embedding space $\mathbb{R}^d$, and a text encoder $T(\cdot)$ which maps the input text $t$ into $\mathbb{R}^d$. We can get an image embedding matrix $\mathbf{I} = [I(x_1), \ldots, I(x_N)]$ and a text embedding matrix $\mathbf{T} = [T(t_1), \ldots, T(t_N)]$, where $\mathbf{I}, \mathbf{T} \in \mathbb{R}^{N \times d}$. The model is trained to maximize the similarity between the embeddings of matching image and text pairs.

$$\min_{I,T} \left[ H(\sigma(\tfrac{\mathbf{I} \cdot \mathbf{T}^\top}{\tau}), \mathbf{Y}) + H(\sigma(\tfrac{\mathbf{T} \cdot \mathbf{I}^\top}{\tau}), \mathbf{Y}) \right], \tag{1}$$

where $\sigma$ is the softmax operation applied in each row , $H(\cdot, \cdot)$ is the cross-entropy function $H(p, q) = -\sum_i p(i) \log q(i)$, and $\tau$ is a learnable temperature coefficient. After training, VLMs become zero-shot classifiers by computing dot product between embeddings of input image $x_0$ and candidate text $t_i \in \mathcal{T}_0$, where $\mathcal{T}_0 = \{t_i|i = 1, \ldots, K\}$ is the set of $K$ text combining the $i$-th category names with prompt texts, the zero-shot classification probability of class $k$ given $x_0$ is $p(k|x_0) = \sigma(\tfrac{I(x_0) \cdot \mathbf{T}_0^\top}{\tau})_k$, where $\mathbf{T}_0 = [T(t_1), \ldots, T(t_K)]$ is the embedding matrix of $\mathcal{T}_0$.

## 2.2 VLMs in Concept Bottleneck Models

Given an input image $x_0$, a CBM predicts concept score $s_c$ for each human-readable concept $c$ via a shared neural network $f(\cdot)$, and get bottleneck concept scores $\mathbf{s} = [s_0, s_1, \ldots, s_C]$. $C$ is the number of predefined concepts. The final model decision is obtained by multiplication with sparse weight matrix $\mathbf{W} \in \mathbb{R}^{C \times K}$ for a $K$-classification problem, the prediction is:

$$\mathbf{p}(k|x_0) = \sigma(\mathbf{W}^\top \cdot \mathbf{s})_k, \text{ where } \mathbf{s} = f(x_0). \tag{2}$$

This prediction score of class $k$, denoted as $v_k$, can be interpreted as $v_k = \sum \mathbf{s}_i \times \mathbf{W}_{ik}$, where $i$ is the index of concepts. Recently, LaBo(Yang et al., 2022) and Lable-Free CBM(Oikarinen & Nguyen, 2023) combine VLMs with CBM to reduce the exhausting labeling effort of $f(\cdot)$. They directly use image encoder $I$ to replace $f$ and get a set of concept text $\mathcal{T}_c$ from LLMs. In this case, the bottleneck concept scores $\mathbf{s}$ are obtained as:

$$\mathbf{s} = \frac{I(x_0) \cdot \mathbf{T}_c^\top}{\tau} \tag{3}$$

where $\mathbf{T}_c = [T(t_1), \ldots, T(t_C)]$ is the embedding matrix of $\mathcal{T}_c$. By this means, the label effort issue is addressed to some extent, and the model performance seems to be improved with respect to the increased number of concepts.

**Prevent information loss (our motivation)** In this paper, we claim that there might be toxic, non-visual, task-irrelevant, or non-factual descriptions in $\mathcal{T}_c$. Moreover, the model might be biased due to the VLMs' module $I$ and $T$ generalization performance in real-world application datasets remains to be explored, leading to potentially biased bottleneck scores $\mathbf{s}$. For better interpretability, we intend to directly obtain a visual concept embedding matrix $\mathbf{E}$ to compute bottleneck scores $\mathbf{s} = \frac{I(x_0) \cdot \mathbf{E}^\top}{\tau}$.

## 3 Concept Decomposition Vector

A concept decomposition vector (CDV), denoted as $\mathbf{e}$, is a vector in the VLM latent space $\mathbb{R}^d$, which captures some key visual concepts that distinguish a class from others. We can combine multiple concepts into a concept matrix $\mathbf{E} = [\mathbf{e}_1, \mathbf{e}_2, \ldots, \mathbf{e}_n]$. Its workflow is shown in the upper part of Figure 3, where both encoders $I$ and $T$ are frozen. With well-trained $\mathbf{E}$, we can perform Decomposed Concept Bottleneck Models (DCBM) by taking place $\mathbf{T}_c$ in Eq 3 for inherent interpretable classification. Section3.1 introduces the quintuple notion of concept and concept-sample distribution, while Section 3.2 outlines how to learn CDVs from a given training dataset. To make each $\mathbf{e}$ as well as its representing concept human-understandable, cross-modality interpretation is performed as described in Section 3.3.

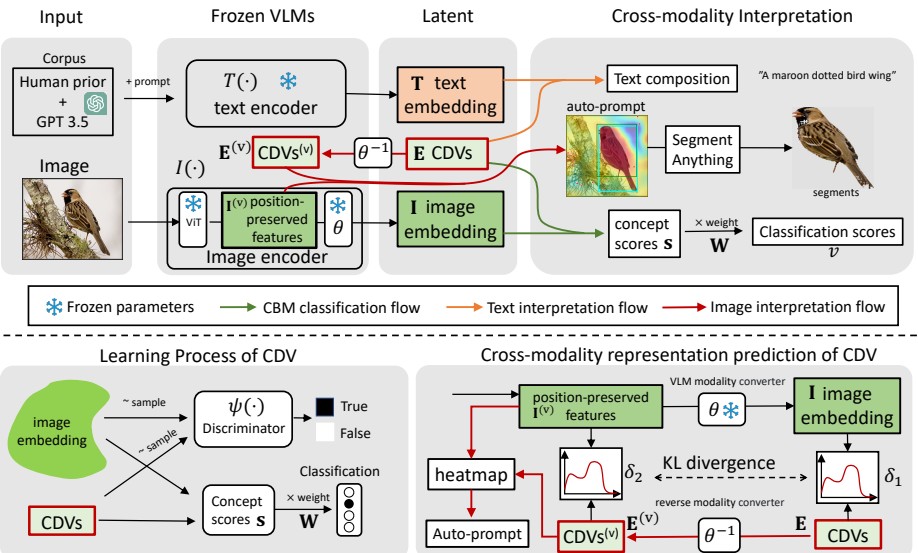

Figure 3: The upper part is the workflow of well-trained CDVs, where cross-modality interpretation is performed following the workflows in three colors. The lower left part is the adversarial training process of CDV. The lower right part is the learning process of cross-modality representation predictor (reverse modality converter) for CDV interpretation in image.

## 3.1 QUINTUPLE NOTION OF CONCEPT AND CONCEPT-SAMPLE DISTRIBUTION

**Definition 1** (Quintuple notion of concept). Each $\mathbf{e}$ is assigned to one category $a$ with a scalar weight $w \in \mathbb{R}$, representing the concept is decomposed from the category. As concepts are mental objects, to let someone realize the concept, a set of image patches $\mathbb{I}$ and a set of text phrases $\mathbb{T}$ needs to be exhibited at the same time. Therefore, the concept represented by CDV $\mathbf{e}$ is denoted as a quintuple so the concept matrix $\mathbf{E}$ contains a set of decomposed visual concepts:

$$\mathcal{E} = \{(\mathbf{e}_i, a_i, w_i, \mathbb{T}_i, \mathbb{I}_i)\}_{i=1}^N. \tag{4}$$

The CDV $\mathbf{e}$, assignment $a$, and weight $w$ are determined in Section 3.2. The image set $\mathbb{I}$ and text set $\mathbb{T}$ of Eq. 4 are determined given CDV $\mathbf{e}$ in Section 3.3. To determine the image set $\mathbb{I}$ and text set $\mathbb{T}$ given CDV $\mathbf{e}$, we need another definition of the concept and sample relationship.

**Definition 2** (Concept-sample distribution). Given a sample set $\mathcal{Z} = \{\mathbf{z}_1, \mathbf{z}_2, \ldots, \mathbf{z}_n\}$ and a concept embedding $\mathbf{e} \in \mathbb{R}^d$, the concept-sample distribution (CSD) is defined as a categorical distribution over the sample set $\mathcal{Z}$ with following probability density function:

$$\delta(k; \mathbf{e}, \mathcal{Z}) = \frac{\exp(\mathbf{e} \cdot \mathbf{z}_k)}{\sum_{\mathbf{z} \in \mathcal{Z}} \exp(\mathbf{e} \cdot \mathbf{z})}, \tag{5}$$

where $\mathcal{Z}$ can either be a text set or an image set. For convenience, we denoted CSD as $\delta(\mathbf{e}, \mathcal{Z})$.

**Proposition 1** The pretraining task of VLMs (Radford et al., 2021), contrastive image-text matching, is to minimize two concept-sample distributions with a shared concept $\mathbf{e}_i$ between different modalities sample set $\mathcal{I}$ and $\mathcal{T}$ given image-text pair $(x_i, t_i)$. Formally, Eq 1 is equivalent to the following objective:

$$\min_{\mathbf{I}, \mathbf{T}} \sum_i^N \left[ \mathrm{KL}(\mathbf{Y}_i \| \delta(\mathbf{e}_i, \mathcal{T})) + \mathrm{KL}(\mathbf{Y}_i \| \delta(\mathbf{e}_i, \mathcal{X})) \right]. \tag{6}$$

The proof is trivial by setting the concept embedding $\mathbf{e}$ as the text embedding $\mathbf{t}_l$ and the sample set $\mathcal{Z}$ as the image embedding $\mathbf{x}_l$ in Eq. 5. This motivates us to use the concept-sample distribution as a learning objective to train a concept embedding in arbitrary latent space.

## 3.2 LEARNING PROCESS OF CONCEPT DECOMPOSED VECTOR

**Initialization.** Given training image dataset with labels $\mathcal{D} = (x_i, y_i)$. we calculate mean $\mu_{\mathcal{X}}$ and variance $\sigma_{\mathcal{X}}$ of image features. Let $C$ be the number of CDVs, $\mathcal{C}$ be a categorical distribution with

equal probability $\bar{p}$, and $\mathcal{U}$ be a uniform distribution. The first three terms of quintuple are initialized as $\mathcal{E} = \{(e_i, a_i, w_i) | e_i \sim \mathcal{N}(\mu_{\mathcal{X}}, \sigma_{\mathcal{X}}), a_i \sim \mathcal{C}(\bar{p}), w_i \sim \mathcal{U}(0,1)\}$. Then we get concept matrix $\mathbf{E}$, and sparse weight matrix $\mathbf{W}$ with $w_i$ as elements on the one-hot embedding of $a_i$.

As we hope that the CDV itself represents a visual concept, we constrain the distribution of CDVs to be consistent with the visual concepts that appeared in the training set. To achieve this, we apply adversarial training to learn CDVs as shown in Figure 3 lower left part. There are two steps in each iteration:

**Step one (train discriminator).** A random initialized 3-layer neural network with non-linear activation $\psi(\cdot)$ acts as a discriminator to tell CDV $\mathbf{e}$ from real image feature $I(x_i)$. In each iteration, we first sample a batch of CDVs $\{\mathbf{e}_i\}$ and a batch of image features $\{\mathbf{z}_i | \mathbf{z}_i = I(x_i)\}$. Then we calculate the loss of discriminator $\mathcal{L}_\psi$ as follows:

$$\mathcal{L}_D = -\frac{1}{m} \sum_{i=1}^{m} \left[ \log \psi(\mathbf{e}_i + \epsilon) + \log(1 - \psi(\mathbf{z}_i + \epsilon)) \right] \tag{7}$$

**Step two (train CDVs).** We use the discriminator to train CDVs to be indistinguishable from real image features. At the same time, we perform interpretable classification by taking $\mathbf{E}$ into Eq 3. Jointly train $\mathbf{E}$ and $\mathbf{W}$ with negative log-likelihood loss. The final loss of step two $\mathcal{L}_{CDV}$ is defined as follows:

$$\mathcal{L}_{CDV} = \underbrace{-\frac{1}{m} \sum_{i=1}^{m} \log \psi(\mathbf{e}_i + \epsilon)}_{\text{discriminator loss}} + \underbrace{\frac{1}{|\mathcal{X}|} \sum y_i \log(\sigma(\frac{I(x_i) \cdot \mathbf{E}^\top}{\sqrt{\eta}} \cdot w^*))}_{\text{classification loss}} + \underbrace{\mathcal{R}(\mathbf{E})}_{\text{regularizer}}. \tag{8}$$

$\mathcal{R}(\cdot)$ is a regularizer to constrain the $\mathbf{E}$ to be as orthogonal as possible.

### 3.3 CROSS-MODALITY INTERPRETATION OF CDV

**Language comprehension via text composition.** Given a CDV $\mathbf{e}$, the set of text $\mathbb{T}$ is sampled from the CSD $\delta(\mathbf{e}, \mathcal{T}_c)$. $\mathcal{T}_c$ is constructed with two strategies with the aid of GPT3.5(Brown et al., 2020): (1) *category-related sentences*: for general category names that LLMs are familiar with, use a prompt combined with the category name for general image label-free descriptions following (Oikarinen & Nguyen, 2023) to get more visual information description. Then $\mathbb{T} = \{t_i | t_i \sim \delta(\mathbf{e}, \mathcal{T}_c)\}$. (2) *category-independent words*: for fine-grained names that LLMs are unfamiliar with, use human prior knowledge to get category-independent words $\mathcal{T}_p$ for the primitive concepts, e.g. colors, shapes. Then $\mathbb{T} = \bigcup_p \{t_i | t_i \sim \delta(\mathbf{e}, \mathcal{T}_p)\}$, and a text composition is optional two organize the words into a sentence by predefined rules.

**Vision comprehension via auto-prompt segmentation.** In VLMs, positional information of images is lost after being embedded by $I(\cdot)$, which makes it difficult to locate the decomposed concept in the image. To address this, we view $I$ as two parts: a ViT that outputs position-preserved embedding $\mathbf{I}^{(v)}$ and a modality converter(Kim et al., 2021) $\theta$ that map $\mathbf{I}^{(v)}$ to $\mathbf{I}$. Each sample in $\mathbf{I}^{(v)}$ includes $1 + L \times L$ tokens. Then we train a neural network $\theta^{-1}$ to act as a reverse modality converter to predict the concept representation of $\mathbf{e}$ in ViT output space. Motivated by Eq 5, the concept representation in another embedding space can be learned by minimizing the KL divergence between the CSDs of the same concept across different modalities. Let $\delta_1 = \delta(\mathbf{e}, \mathbf{I})$ and $\delta_2 = \delta\left(\theta^{-1}(\mathbf{e}), \mathbf{I}_0^{(v)}\right)$, the training objective of $\theta^{-1}$, which is also shown in Figure 3 lower right part, is:

$$\min_{\theta^{-1}} [\mathrm{KL}(\delta_1 \| \delta_2) + \mathrm{KL}(\delta_2 \| \delta_1)]. \tag{9}$$

After training, we get $\mathbf{E}^{(v)} = \{\theta^{-1}(\mathbf{e}_1), \theta^{-1}(\mathbf{e}_2), \ldots, \theta^{-1}(\mathbf{e}_C)\}$. Then similarity heatmaps can be calculated between the image embeddings and CDVs $I_{L \times L}^{(v)}$, generating a bounding box of high similarity area. then we crop the box and success location of decomposed concepts on image regions. The bounding box can be further fed into SAM(Kirillov et al., 2023) as auto-prompts to get finer segments.

## 4 EXPERIMENT

The experiments are conducted for the following goals: (1) to compare the classification performance of using CDV rather than the text concepts from LLM across multiple image domains. (2) to

evaluate the quality of cross-modality interpretation from different aspects, including concept level, sample level, and class level. (3) to compare the interpretability with other CBM methods in terms of both automatic evaluation and human evaluation.

## 4.1 CLASSIFICATION PERFORMANCE ANALYSIS

**Datasets.** (1) **Natural** images to evaluate general classification performance, using the well-known image dataset CIFAR-100 (Krizhevsky et al., 2009); (2) **Semantic** images with clear concepts in their class names, including DTD (Cimpoi et al., 2014), a texture dataset containing 47 human-recognizable textures, and UCF101 (Soomro et al., 2012), a human actions dataset with 101 human actions; (3) **Fine-grained** images that require some additional knowledge. CUB-200-2011(Wah et al., 2011) without cropping, a bird dataset containing 200 different classes of birds, and FGVC-Aircraft(Maji et al., 2013), an aircraft dataset containing 100 different classes; (4) **Specialized** images from real-world applications with special camera. EuroSAT (Helber et al., 2019), a satellite remote sensing image dataset containing 10 kinds of land use types, HAM10000 (Tschandl et al., 2018), a medical image dataset containing 7 kinds of skin diseases, Diabetic Retinopathy (Karthik, 2019), a dataset containing 5 types of diabetic retinopathy, and Keratitis(Fang et al., 2020), a slit-lamp image dataset containing 4 kinds of infectious keratitis diseases.

**Baseline.** Four methods are choosen, including black-box linear probe(He et al., 2022), sparse linear probe(Wong et al., 2021) for sparse layers have been demonstrated to be more interpretable(Oikarinen & Nguyen, 2023), Label-free CBM(Oikarinen & Nguyen, 2023), and LaBo(Yang et al., 2022). None of these methods alter the image encoder parameters. The performance of linear probe serves as the benchmark for interpretable methods. For the choice of VLM, all methods employ the same pretrained CLIP model with ViT-B/16 and ViT-L/14 as image backbones. The same *train/dev/test* split with Yang et al. (2022) is setted and select the best validation performance on *dev*, reporting the average classification accuracy of five runs with random seeds 41-45. The classification results on *test* are shown in Table 1(*dev* results are in Appendix).

Table 1: Classification Accuracy of four interpretable methods on all *test* sets

| Dataset Type | | Natural | Semantic | | Fine-grained | | Specialized | | | |
|---|---|---|---|---|---|---|---|---|---|---|
| Dataset Name | | CIFAR100 | DTD | UCF101 | CUB | Aircraft | EuroSAT | HAM10000 | DR | Kera |
| ViT-B-16 | linear probe* | 75.62% | 77.89% | 86.59 | 77.50% | 52.31% | 96.04% | 80.64% | 53.22% | 68.01% |
| | sparse LP | 59.84% | 74.86% | 80.21% | 63.87% | 42.67% | 92.15% | 76.32% | 50.55% | 62.59% |
| | Label Free CBM | 58.79% | 69.05% | 77.29% | 59.35% | 35.57% | 92.21% | 72.16% | **53.80%** | 49.10% |
| | Labo | 73.93% | 75.18% | **85.67%** | 76.48% | **51.48%** | 93.87% | 80.30% | 47.51% | 50.13% |
| | Ours | **75.37%** | **77.39%** | 85.60% | **77.36%** | 50.25% | **95.44%** | **80.74%** | 52.21% | **67.77%** |
| ViT-L-14 | linear probe* | 80.87% | 81.17% | 90.12% | 84.15% | 62.42% | 97.22% | 80.66% | 55.10% | 67.14% |
| | sparse LP | 74.29% | 80.54% | 87.23% | 81.24% | 58.03% | 95.70% | 78.55% | **53.63%** | 66.66% |
| | Label Free CBM | 46.54% | 66.84% | 74.46% | 56.42% | 29.49% | 74.65% | 70.05% | 53.42% | 44.43% |
| | Labo | 79.62% | 77.30% | **90.11%** | 81.90% | **61.06%** | 95.82% | **81.39%** | 48.48% | 44.44% |
| | Ours | **80.86%** | **81.12%** | 89.57% | **83.95%** | 60.67% | **96.89%** | 81.05% | 52.73% | **67.23%** |

**Compared to linear probe.** Table 1 indicates that DCBM shows comparable performance to the linear probe on most datasets, including natural images, semantic images, and specialized images. This suggests CDV has sufficiently utilized the embedded feature. Moreover, our approach offers the added benefit of interpretability compared to linear-probe by the CBM-like classifier.

**Compared to VLM-based CBM.** DCBM outperforms two VLM-based CBM methods, LaBo and Label-free CBM, on most datasets, particularly for natural images. On semantic images, DCBM improves significantly over both methods on DTD and on UCF101 compared to Label-free CBM, with only a marginal difference of 0.31% in average classification accuracy compared to LaBo. For specialized images, DCBM shows a significant improvement over both methods on EuroSAT and Kera datasets, but performs marginally lower than LaBo by 0.1% on average in HAM10k dataset. However, on the DR dataset, DCBM and LaBo perform worse than Label-free CBM by 1.14%, indicating a potential benefit of utilizing an external CNN feature extractor. On fine-grained images, DCBM significantly outperforms LLM-CBMs on CUB, while trailing LaBo on aircraft, potentially due to non-visual text from LLMs that could compromise interpretability (shown in Appendix).

## 4.2 CROSS-MODALITY INTERPRETATION WITH CDV

We performed three levels of interpretation: concept level, sample level, and class level.[1]

---

[1]All interpretations use ViT-L/14 as the image encoder of CLIP. Refer Appendix A,B,C for more showcases.

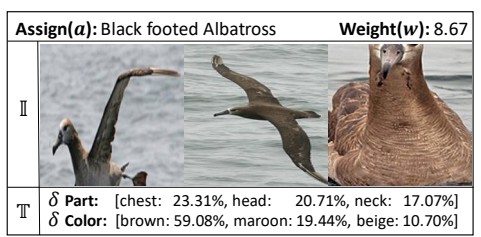

(a) Category-independent words as text descriptions.

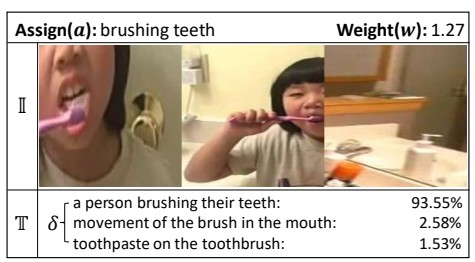

(b) Category-related sentences from LLM as text descriptions.

Figure 4: Example of cross-modality understanding of CDV.

**Concept level interpretation.** Figure 4 illustrates the presentation of two randomly selected CDVs from CUB and UCF101 datasets. For each CDV, we show three image patches that are most suitable for representing the CDV, and some text that is most suitable for representing its semantics using the image-text matching method. In the example from CUB, a concept related to the Black-footed Albatross has a weight of 8.67, and the top three words with the highest probabilities are printed. In this case, we use the category-independent text descriptor. For body parts, 'chest', 'head', and 'neck' have probabilities of 23.31%, 20.71%, and 17.07%, respectively, while for colors, 'brown' and 'marron' have higher probabilities, see Figure 4a. Similarly, we use the same approach for the sample of UCF101. We use the category-related text descriptor in this case. The result shows that 'a person brushing their teeth' has the highest possibility (93.55%) for interpreting the CDV.

**Sample level interpretation.** We illustrate sample-level explanations for DCBM using a randomly selected image from the UCF101 and Kera datasets, see Figure 5.

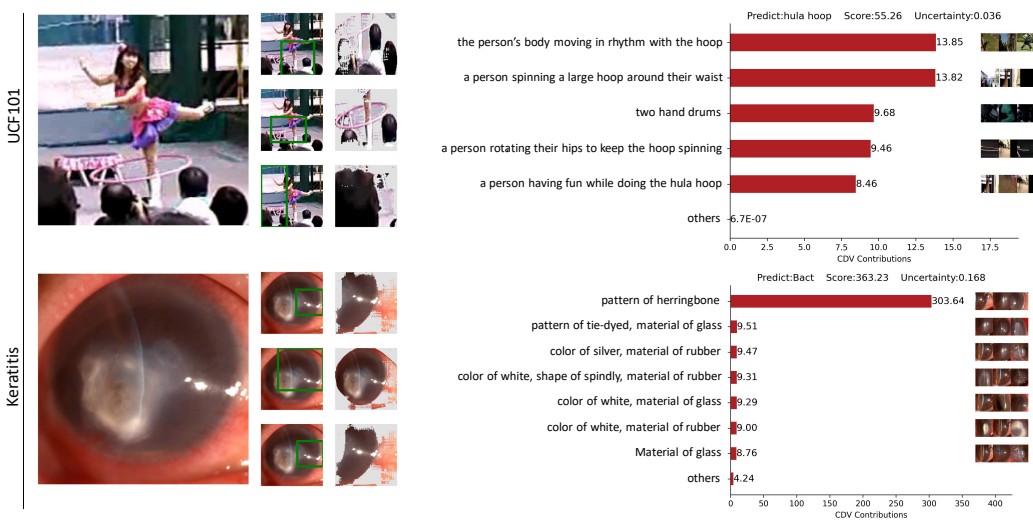

Figure 5: Interpretation of two examples on UCF101 and Keratitis made by our method.

Horizontal bar charts depict the values obtained by multiplying concept scores with their weights, accompanied by corresponding textual explanations and the top three images.

Our interpretable classification successfully identifies crucial visual information, explains it using text, and provides classification scores. We also present corresponding images for each concept, allowing users to examine the concept's accuracy and correspondence to visual information from different perspectives.

**Class level interpretation** CBMs can be explained as a linear combination of interpretable features, where the weights can be regarded as their importance for classification. Fine-grained datasets

demand specific domain knowledge for accurate classification. Cross-modal explanations can help understand these classification challenges.

We created a Sankey diagram to visualize the final layer weights for 'Black-footed Albatross' and 'Carolina Wren' in CUB, see Figure 6. The width of the lines connecting a concept to an output class represents their weight, with only weights greater than 0.05 included. The two birds with similar brown chest and head features may be easily confused. However, the 'Black-footed Albatross' exhibits unique streamlining and herringbone patterns absent in the 'Carolina Wren'. Conversely, the 'Carolina Wren' has distinct beak and spherical features.

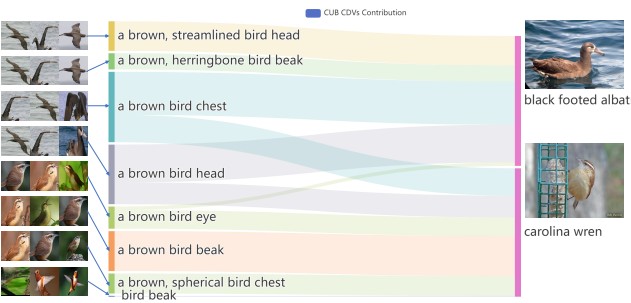

Figure 6: Classification challenge in CUB dataset.

### 4.3 INTERPRETABILITY COMPARISON WITH OTHER CBM MODELS

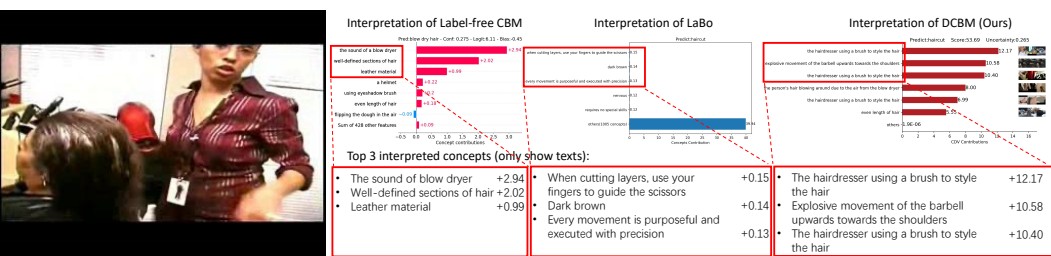

Figure 7: Comparison between LLM-VLM based concept bottleneck models.

An intuitive comparison between DCBM with previous methods is shown in Figure 7. To quantitative evaluation, we conducted a questionnaire on the interpretation to verify which method's interpretation results are more in line with human perception.

**Method.** The evaluation is conducted via an online questionnaire, where 8 cases are randomly sampled from UCF101 and DTD for each test. Images from fine-grained and specialized domain datasets are not chosen because extra knowledge is necessary for their recognition. We compare DCBM with LaBo and Label-free CBM. The testers are asked to rank the three interpretation methods four times for each case according to different questions. Participants are invited for the diverse geographic distribution (Americas, Asia, and Europe) to make our research as representative as possible. In total, we collected ratings from 27 testers from computer science undergraduate/graduate students and medical graduate students.

**Metrics.** Four metrics are considered in human validation: (1) **Groundability** Yang et al. (2022), with the question 'Which interpretation texts are more consistent with the image content'; (2) **Factuality** Yang et al. (2022), with the question 'Which interpretation texts are consistent with the label'; (3) **Meaningfulness** Ghorbani et al. (2019a), with the question 'Which interpretation texts are more semantic' and (4)

Table 2: Definition of evaluation metrics

| Metric | Definition | Range |
|---|---|---|
| Accuracy | $(TP+TN)/(TP+TN+FP+FN)$ | $[0,1]$ |
| Sparsity | $\text{avg}\left(\sum_{i=0}^{n} s_i \times \mathbf{W}_{ik}/v_k\right)$ | $[0,1]$ |
| Groundability | | $[0,1]$ |
| Factuality | $\dfrac{\sum 3 \times N_1 + 2 \times N_2 + 3 \times N_3 + 0 \times N_{n/a}}{3 \times N}$ | $[0,1]$ |
| Meaningful | | $[0,1]$ |
| Fidelity | | $[0,1]$ |

**Fidelity** Velmurugan et al. (2021), with the question 'Which interpretation scores are more supportive to the predictions'. We record the frequencies of rank for each method, where $N_1$ denotes the number of times a method is at the first rank, the same as $N_1$ and $N_2$, and $N_{n/a}$ denotes the number of times that not been incorporated in the ranking. The metric calculations are shown in Table 2, where we also incorporate average **Accuracy** on DTD and UCF101 for evaluating the trade-off

between performance and interpretability. Sparsity is also calculated automatically to evaluate the proportion of the top 5 concept scores to all concepts.

**Result.** According to Figure 8, our method demonstrated a considerable improvement in the six metrics compared to LaBo and Label-free CBM. This shows that our approach is more easily accepted by humans in the context of explanation, and the explanation results are more objective and consistent with human understanding. Regarding the sparsity metric, LaBo's approach exhibits lower values compared to the others. This discrepancy arises from LaBo assigning high weights to all concepts for a single sample. Furthermore, Label-free is very close to DCBM in terms of sparsity and factuality, which is essential in real applications, making users less influenced by the LLM's non-factual outputs. This compelling evidence suggests that our interpretation results are more easily recognizable and comprehensible when compared to the alternatives.

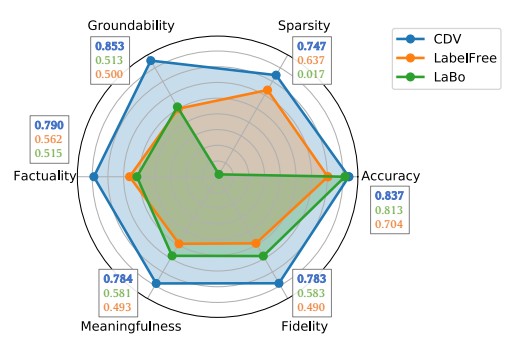

Figure 8: Radar chart of interpretability evaluation comparison in 6 metrics.

### 4.4 ABLATION STUDY

In Appendix E1, we compare the model performance of various methods under different number of concepts. The results indicates that DCBM does not rely on a large number of concepts. In Appendix E2, we study the impact of CDV initialization and the neccesscity of discriminator.

## 5 RELATED WORK

**Concepts-based explanation.** Kim et al. (2018) proposes concept activation vectors, which inspires us that concepts can be represented by vectors, but a similar approach to construct concept vectors requires a large number of annotations. ACE(Ghorbani et al., 2019b) obtains concepts by clustering, and we exploit the evaluation metrics of concept semantics in it. *ante-hoc methods.* Ante-hoc provides reasoned decision processes and is therefore popular for high-risk decisions, but there is a trade-off in interpretation and performance. Concept Bottleneck Models (CBM)(Koh et al., 2020; Zarlenga et al., 2022) is a representative class of models, but requires a large amount of annotation. Recent work(Yuksekgonul et al., 2022; Oikarinen & Nguyen, 2023; Yang et al., 2022) using VLM and LLM has reduced the amount of annotation and also increased effectiveness. For our method, the decision process is ante-hoc and the cross-modality interpretation is post-hoc with CSD. Just as different people have different mental mappings of the same person, we believe that recognizing abstract concepts has a certain uncertainty, so the post-hoc interpretation is represented by the distribution. Other related works are shown in Appendix F.

## 6 DISCUSSION

We propose CDV, which serves dual purposes of cross-modality interpretation and constructing high-performance and interpretable CBMs. CDV, as primitive concepts derived from training data, aligns with human cognitive pathways, thus reducing bias in previous methods. The cross-modality interpretation of CDVs not only enhances people's understanding of important concepts but also holds significant value in real-world applications, such as assisting doctors in identifying important features for diagnosis and generating text descriptions. The quintuple notion of concept and CSD, as a novel theory, offer a new explanation for VLM image-text matching. The success of cross-modality interpretation also validates the hypothesis that CSD can be used to learn concepts in different latent spaces. However, the limitation is that this work has currently only been validated on CLIP, its extensions to other VLMs needs further study. The CNN backbone has not been explored either. In future work, we will explore the application of cross-modality interpretation in specific scenarios and attempt pre-training with more than three modalities using the CSD theory.

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

## A   APPENDIX

You may include other additional sections here.

