# Supplementary Material

## A  Additional demonstrations of concept-level interpretation

In Figure 1, we present additional concept-level interpretations for six datasets, each with two CDVs learned by different models. The results demonstrate that our method can explain the concepts learned by the models for classification in two different modalities, making them easier to understand. Moreover, the images and text for each concept are matched, indicating consistency in cross-modality concept explanations. It is worth noting that for the CUB and HAM10000 datasets, we used category-independent words, while category-related words were used for the other four datasets. Category-independent words are more objective but may also be more simplistic. The choice of text interpretation method depends on the specific use case; when a model is used to differentiate fine-grained images or specialized domain images, we recommend using category-independent words.

## B  Additional demonstrations of sample-level interpretation

In Figure 2, we present additional sample-level interpretations for five datasets, with each dataset visualizing the interpretations for one sample. The results demonstrate that our method can accurately identify the concepts that match the sample image and convert them into consistent visual and textual interpretations. Additionally, our method's sparsity is noteworthy, as only a portion of the concepts are activated for each sample. This makes our method more easily understandable and acceptable for humans. For each sample, we also display the contribution score of the interpretation result to the classification and rank them, which helps users understand the model's behavior and enables interactive feedback on the concepts.

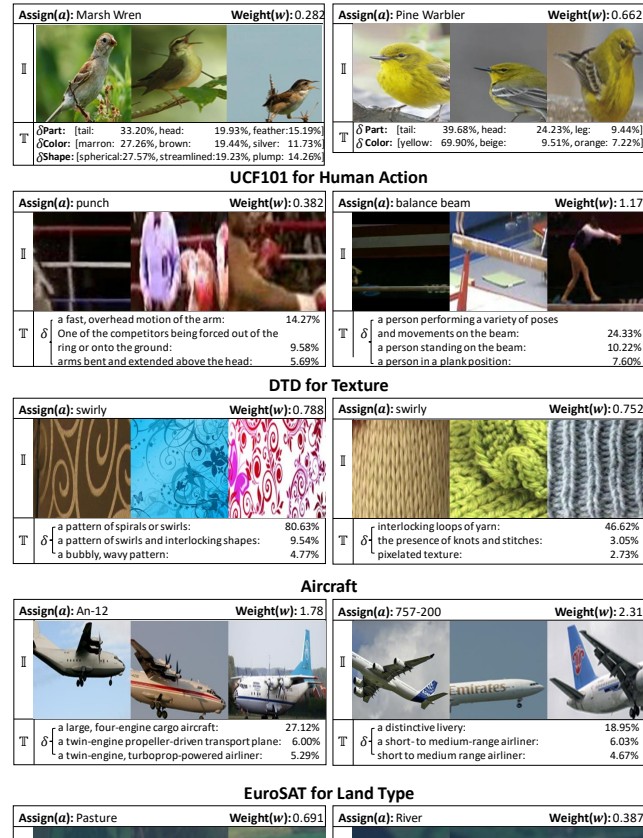
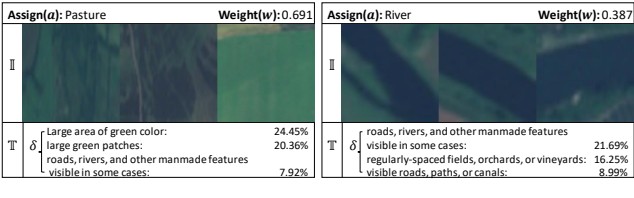
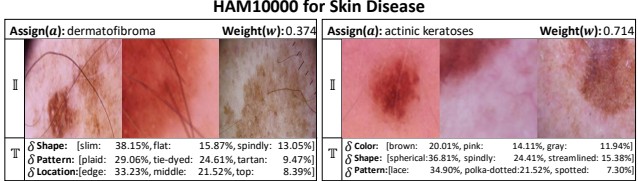

Figure 1: Interpretation for 12 randomly chosen CDVs from 6 datasets.

## C  Additional demonstrations of class-level interpretation

In Figure 3, we use Sankey diagrams to visualize the contributions of CDVs to five other datasets. The Sankey diagrams provide a clear visual representation of which CDVs are more valuable and pivotal for classification when using CDVs in the classification process.

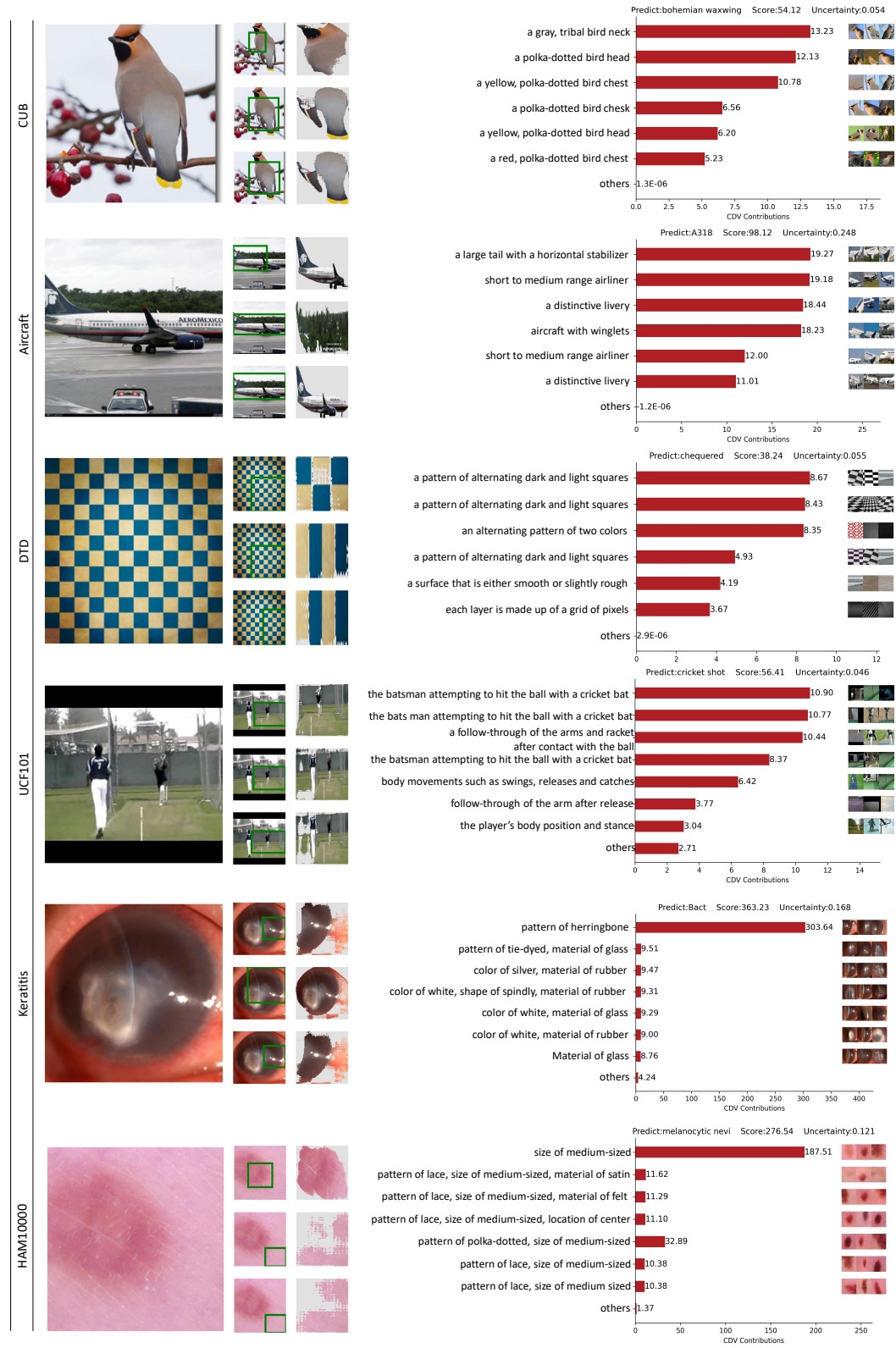

Figure 2: Interpretation for 5 randomly chosen samples with our method.

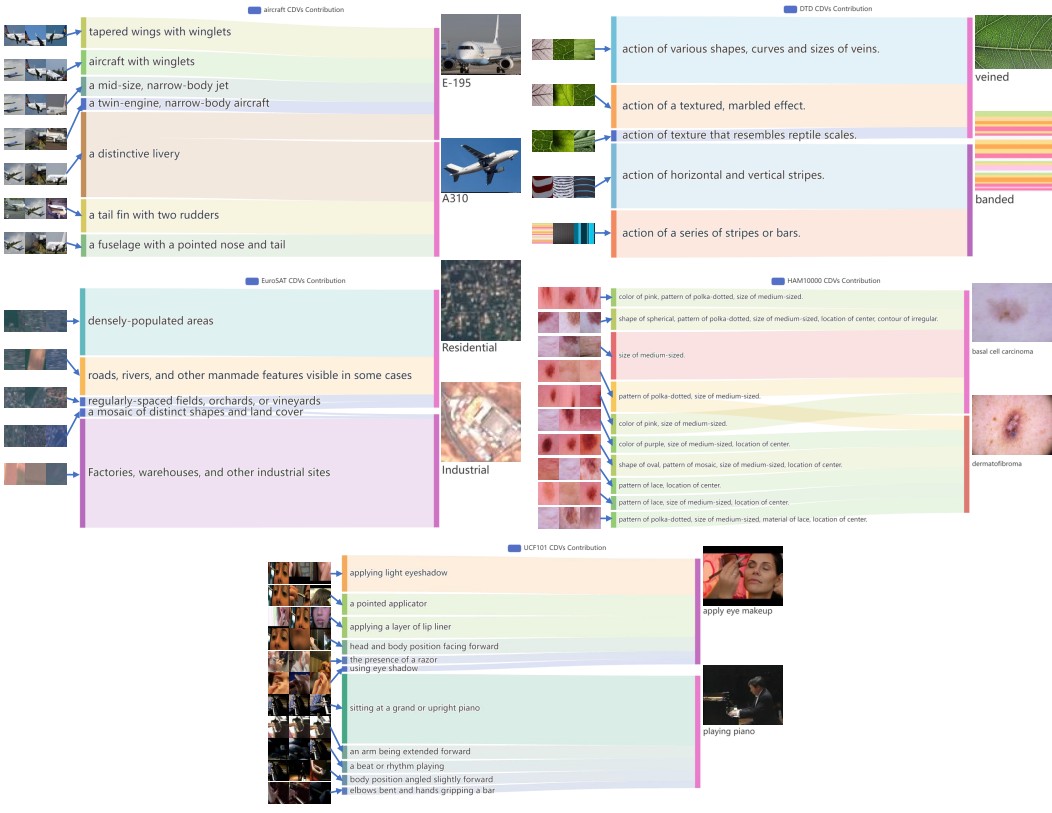

Figure 3: Visualization of the CDVs' contribution to the classes in 5 datasets.

## D Human Evaluation

To verify whether the interpretation of the sample by our method is consistent with human perception, we conducted human-machine experiments. We selected four samples from each of the DTD and UCF101 datasets as test samples (see Figure 4 for examples) because these two datasets are more common and humans can easily determine their features and categories. We defined 4 human experimental metrics as groundability, factuality, meaningful and fidelity. Please refer to Table **??** for the relevant definitions.

During the human evaluation process, we presented the same sample to three different methods(Label-free CBM, LaBo, and ours) for interpretation, and they were then presented to human validators for scoring. To ensure experimental fairness, we chose samples that were correctly predicted by all three methods and anonymized the method names. To simplify the scoring process for the validators, we adopted a ranking-based scoring system in which the evaluators only needed to rank the sample's relative strength/weakness in a particular aspect. The final scores were computed by aggregating the weights assigned to the rankings (see Table **??** for the calculation method). The resulting scores across different aspects for each method are presented in Table 1.

| metrics | accuracy | sparsity | groundability | factuality | meaningful | fidelity |
|---|---|---|---|---|---|---|
| Ours | **0.8373** | **0.7465** | **0.8533** | **0.7900** | **0.7842** | **0.7829** |
| Label-free CBM | 0.7036 | 0.6370 | 0.5000 | 0.5617 | 0.4929 | 0.4896 |
| Labo | 0.8126 | 0.0174 | 0.5133 | 0.5154 | 0.5813 | 0.5829 |

Table 1: Complete results of human evaluation.

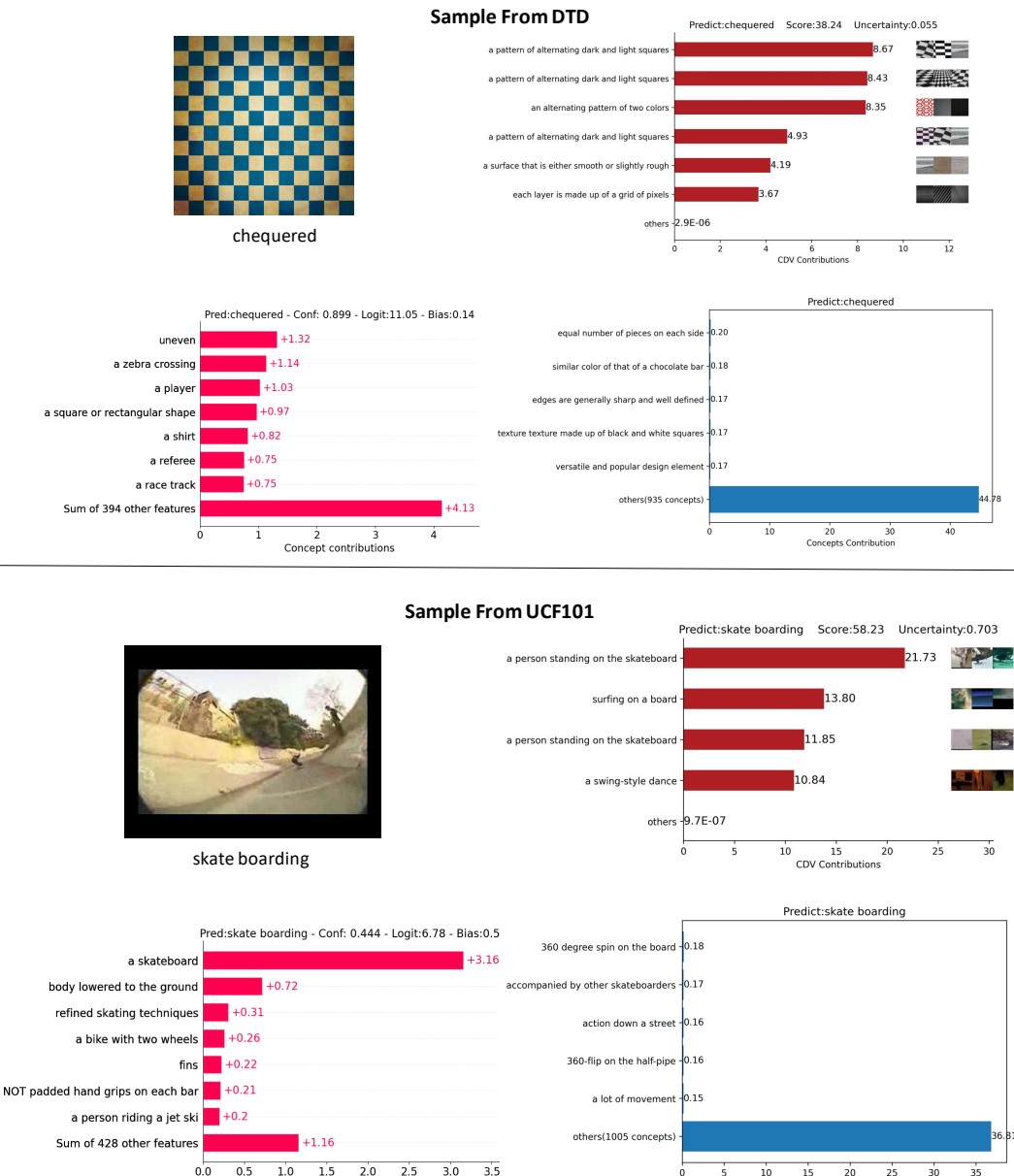

Figure 4: Two samples used for human evaluation.

# E    Ablation Study

## E.1    Number of CDVs

In order to compare the model performance of various methods under different number of concepts, we divided the concept quantity into five levels, namely [3, 5, 7, 10, 20], and the corresponding specific concept quantity is its product with the number of categories. We conducted experiments on different number of concepts. Table 2 shows the accuracy on dev for each model when using different number of concepts, and shows it in the form of line chart(see Figure 5). As the number of concepts increases, the performance of the model generally shows an upward trend. However, unlike other methods, our method still performs well when there are fewer concepts, and its performance does not significantly decrease when compared to a large number of concepts. This indicates that our method does not rely on a large number of concepts and can grasp key concepts when interpreting,

Table 2: Accuracy on dev under different concept quantities

| Dataset Type | | Natural | Semantic | | Fine-grained | | Specialized | | | |
|---|---|---|---|---|---|---|---|---|---|---|
| Dataset Name | | CIFAR100 | DTD | UCF101 | CUB | Aircraft | EuroSAT | HAM10000 | DR | Kera |
| Label Free CBM | 3 | 36.64% | 59.22% | 84.25% | 54.15% | 30.90% | 63.41% | 68.00% | 53.31% | 51.15% |
| | 5 | 45.09% | 63.21% | 86.56% | 57.05% | 34.26% | 75.85% | 69.60% | 53.93% | 49.51% |
| | 7 | 49.33% | 65.25% | 87.20% | 58.30% | 35.43% | 84.11% | 69.10% | 53.24% | 50.49% |
| | 10 | 52.33% | 67.99% | 89.46% | 59.60% | 36.42% | 87.78% | 71.50% | 52.69% | 48.52% |
| | 20 | 54.50% | 67.73% | 89.88% | 60.20% | 37.26% | 90.78% | 72.30% | 56.62% | 51.80% |
| LaBo | 3 | 78.85% | 72.61% | 96.47% | 79.90% | 57.55% | 89.85% | 71.30% | 45.23% | 52.65% |
| | 5 | 79.10% | 75.27% | 96.68% | 80.65% | 58.96% | 93.15% | 73.90% | 47.10% | 52.46% |
| | 7 | 79.71% | 74.91% | 97.10% | 80.45% | 60.34% | 94.19% | 75.40% | 49.93% | 53.79% |
| | 10 | 79.77% | 75.35% | 97.21% | 81.00% | 60.85% | 94.74% | 76.10% | 50.55% | 51.33% |
| | 20 | 80.03% | 76.68% | 97.42% | 81.10% | 61.45% | 95.63% | 79.60% | 49.72% | 53.41% |
| Ours | 3 | 78.37% | 75.94% | 92.05% | 80.11% | 59.53% | 96.80% | 83.12% | 57.79% | 70.69% |
| | 5 | 81.25% | 79.02% | 97.98% | 82.62% | 61.45% | 96.79% | 83.60% | 57.90% | 70.23% |
| | 7 | 81.68% | 79.47% | 98.12% | 83.69% | 62.18% | 96.96% | 83.58% | 58.52% | 71.15% |
| | 10 | 81.99% | 79.43% | 98.19% | 83.97% | 62.39% | 96.99% | 83.78% | 58.54% | 71.48% |
| | 20 | 82.17% | 79.79% | 98.36% | 84.26% | 62.68% | 96.96% | 83.72% | 58.58% | 71.61% |

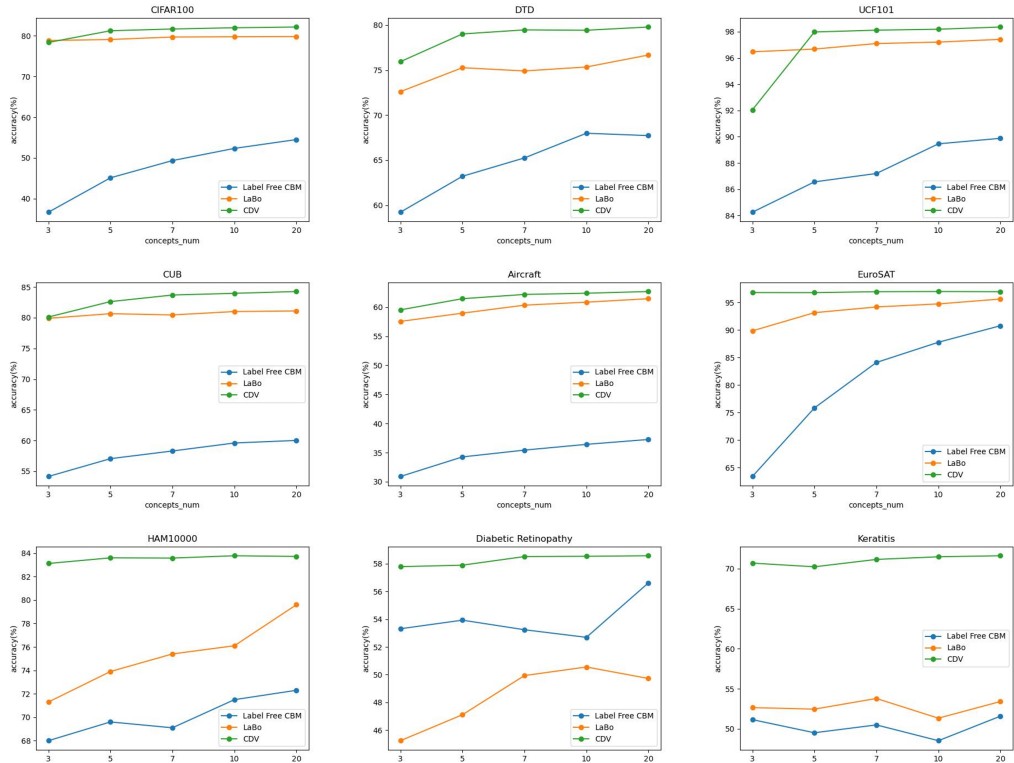

Figure 5: The impact of concept quantity on model performance. The horizontal axis represents the number of concepts. We have selected five levels [3, 5, 7, 10, 20], and their product with the number of categories is the total number of concepts selected in the corresponding dataset. The vertical axis represents the accuracy of the model.

providing more sparse explanations. It should be noted that when the number of concepts on the UCF101 dataset is three times the number of categories, the performance of the model is significantly decrease because of the need for a more detailed description of the action recognition task. Other methods obtain concepts with high-level semantic information from LLM, but we start from the most original concepts. When the number of concepts is extremely small, it is difficult to identify actions with a sparse linear layer.

initialization and discriminator: Our initial strategy is as follows Given training image dataset with labels $\mathcal{D} = (x_i, y_i)$. we calculate mean $\mu_\mathcal{X}$ and variance $\sigma_\mathcal{X}$ of image features. Let $C$ be

| Method | Dataset | | | | | | | | |
|---|---|---|---|---|---|---|---|---|---|
| | CIFAR100 | DTD | UCF101 | CUB | Aifcraft | EuroSAT | HAM10000 | DR | Keratitis |
| linear probe | 75.62% | 77.89% | 86.59% | 77.50% | 52.31% | 96.04% | 80.64% | 53.22% | 68.01% |
| DCBM(w/o $\Phi$) | 75.62% | 77.62% | 86.10% | 77.24% | 51.88% | 96.04% | 81.03% | 52.90% | 67.23% |
| DCBM(random) | 75.76% | 78.36% | 86.30% | 78.28% | 52.51% | 95.97% | 79.98% | 53.19% | 67.47% |
| DCBM(w) | 75.37% | 77.39% | 85.60% | 77.36% | 50.25% | 95.44% | 80.74% | 52.21% | 67.77% |

the number of CDVs and randomly assign a category $a_i \sim \mathcal{C}(\bar{p})$ to each CDV., where $\mathcal{C}$ is a categorical distribution with equal probability $\bar{p}$. The weight $w_i$ is initialized as $w_i \sim \mathcal{U}(0,1)$, where $\mathcal{U}$ is a uniform distribution. The first three terms of quintuple are initialized as $\mathcal{E} = (e_i, a_i, w_i)|e_i \sim \mathcal{N}(\mu_\mathcal{X}, \sigma_\mathcal{X}), a_i \sim \mathcal{C}(\bar{p}), w_i \sim \mathcal{U}(0,1)$. Then we get concept matrix $\mathbf{E}$, and sparse weight matrix $\mathbf{W}$ with $w_i$ as elements on the one-hot embedding of $a_i$.

### E.2 Discriminator and CDV initialization

The discriminator $\Phi$ is applied to ensure the visual semantics of the learned concept vector. Technically, adversarial training is efficient to minimize the distance between two distributions With $\Phi$ and adversarial training, CDVs are ensured to be the real semantics in the training sets.

The initialization of CDV ($e_i$) may influent the performance and interpretation of CDV. We conducted the ablation results to investigate the absence of initiation (DCBM random) and the absence of discriminator (DCBM w.o $\Phi$) in the following table. The ablation study is conducted on ViT-B-16 and ViT-L-14 on all mentioned datasets. We can find that the performance of (DCBM w.o $\Phi$) and (DCBM random) are generally higher than the full model, and even slightly higher than linear probe, which might be the benefit of added parameters.

We also directly visualize their impact on the interpretation of CDV via t-SNE and visualization in Figure 2 and 3 of the one-page pdf. the technique details of modality converter ($\theta$) and reverse modality converter ($\theta^{-1}$). The modality converter ($\theta$) is actually a part of the pre-trained VLMs. In CLIP, it is a frozen 1-layer fully connected neuron network to align image features to text embeddings. The reverse modality converter ($\theta^{-1}$) is proposed in this work, which aims to predict the concept embedding in the intermediate layer. $\theta^{-1}$ is a small 3-layer MLP taking 512-dim vectors as input and output 768-dim vectors, trained with the loss of Eq (10). For the ablation study of Eq (10), we sort the test samples for each concept embedding and compare the sequence rank consistency with the Spearman coefficient. The results are shown below.

## F  Related work

**Concept-based explanations for image classification.** The concept-based explanation is a form of explaining deep learning models that use high-level human-understandable concepts rather than low-level pixels or heatmaps. The important concepts for classification are presented by image segments or readable texts.

*post-hoc methods.* Post-hoc methods try to explain a well-trained black-box model. TCAV**?** proposes concept activation vectors, which inspires us that concepts can be represented by vectors, but a similar approach to construct concept vectors requires a large number of annotations. ACE**?** obtains concepts by clustering, and we exploit the evaluation metrics of concept semantics in it.

*ante-hoc methods.* Ante-hoc provides reasoned decision processes and is therefore popular for high-risk decisions, but there is a trade-off in interpretation and performance. Concept Bottleneck Models (CBM)**??** is a representative class of models, but requires a large amount of annotation. Recent work**???** using VLM and LLM has reduced the amount of annotation and also increased effectiveness. For our method, the decision process is ante-hoc and the cross-modality interpretation is post-hoc with CSD. Just as different people have different mental mappings of the same person, we believe that recognizing abstract concepts has a certain uncertainty, so the post-hoc interpretation is represented by the distribution.

**Visual Language Models and Large Language Models.** Visual Language Models (VLMs) are a series of models that can understand and generate both images and text, including CLIP**?**, BLIP**?**, BLIP2**?**, and GLIP**?**, etc.

*Prompt engineering of VLMs.* CLIP**?** shows selecting a better prompt for VLM can significantly improve performance in many situations. Prompt design is a rapidly evolving research area and has garnered substantial recent attention and activity**??????**. Prompt engineering demonstrates strong classification performance in the few-shot case; however, the process of converting prompts into representations for learning poses challenges in explaining the model's behavior during classification. In contrast, our approach excels in this aspect and provides a detailed understanding of the model's behavior when completing classification tasks.

*Adapters for VLMs and LLMs.* Adapters are a type of method that add layers and parameters to acquire fresh knowledge from new data domains without altering the original model parameters**???**. Although our approach and the adapter method share the use of variable model parameters, our approach emphasizes elucidating the model's behavior while preserving its original capabilities, whereas the adapter method is solely focused on performance enhancement for a specific task.

*Risks and biases in VLMs and LLMs.* There are several works about the risks and biases of visual language models and large language models. A modality gap in VLM is reported by **?**, which can be explained with the concept-sample distribution proposed in this paper. A concept association bias is also reported by **?**, causing false answers in VQA tasks when two concepts appear in the image at the same time. Our work decomposed the concept in the image, we will examine if this decomposed concept can be used to avoid such association bias in future works. As for LLM, the ChatGPT reported that there are concerning patterns where specific entities (e.g., certain races) are targeted more than others **?**. These biases are accumulated step by step in LLM-VLM-CBM workflows. Our work is aim to relieve the bias by directly use primitive visual concepts for classification.

**Concept compositionality and concepts in visual-language models** Concept compositionality is one of the shared perspectives between psychologists and neural scientists. It suggests human minds might employ a language-like system for combining and recombining simple concepts to form more complex thoughts **?**. The compositionality is also regarded as one of the sources of the human brain's few-shot learning and generalization ability **?** which emerges in visual language models (VLMs) **?**. Recent research tests the compositionality of pre-trained VLMs and emphasizes the acquisition of primitive concepts is necessary for interpretation **?**. In this paper, the necessity is admitted but we further claim the primitive concepts for interpretation should come from images rather than languages to increase interpretability.

# G   Adversarial Training

We first initialize the CDVs by randomly assigning a category $a_i \sim \mathcal{C}(\bar{p})$ to each CDV, where $\mathcal{C}$ is a categorical distribution with equal probability $\bar{p}$, and set the weight of CDVs to follow a uniform distribution. Then we get concept matrix $\mathbf{E}$, and sparse weight matrix $\mathbf{W}$ with $w_i$ as elements on the one-hot embedding of $a_i$. As we hope that the CDV itself represents a visual concept, we constrain the distribution of CDVs to be consistent with the visual concepts that appeared in the training set. To achieve this, we apply adversarial training to learn CDVs.

---

**Algorithm 1** Adversarial Training of Concept Decomposition Vector (CDV)

---

**Input:** Real image representation distribution $\mathcal{X}_z$, CDVs $E$, discriminator $D$, number of training iterations $T$, batch size $m$, learning rate $\alpha$
**Output:** Learned CDV $E$.
1: Initialize CDVs $E$ and discriminator $D$.
2: **for** $t = 1$ to $T$ **do**
3:      // Train discriminator
4:      Sample $m$ real image representations $z_1, z_2, \ldots, z_m$ from $\mathcal{X}_z$
5:      Sample $m$ CDV representations $e_1, e_2, \ldots, e_m$ from CDV $E$
6:      Compute discriminator loss: $\mathcal{L}_D$
7:      Update discriminator parameters: $\theta_D \leftarrow \theta_D - \alpha \cdot \nabla_{\theta_D} L_D$
8:      // Train CDVs
9:      Compute CDVs loss: $\mathcal{L}_{CDV}$
10:     Update CDVs parameters: $E \leftarrow E - \alpha \cdot \nabla_{\theta_E} \mathcal{L}_{CDV}$
$\epsilon \sim \mathcal{N}(0, \mathbf{I})$ is a random noise and $R(E) = \langle E, E^\top \rangle - \mathbf{I}$ is a regularizer.

---

### G.1 The prompt and example corpus

The original CLIP paper mentions that applying prompts to the text can influence the process of image-text matching. Applying category information to the text prompts can alter the distribution of textual features and thus affect the matching between the text and image. In our method, we also apply prompts for VLM. When using category-independent words as textual descriptors, prompts are necessary as the vocabulary does not contain information about the image type. For each word attribute, we use prompts in the form of "A photo of {category_name} with a {attribute} of {attribute value}." For example, "A photo of skin disease with a color of red." Using these prompts can effectively improve the model's accuracy in recognizing attributes.

## H  Proof of proposition 1

Image text matching (ITM) is the common training objective of VLMs that maps image representation into a language concept embedding space. It is important to understand how it works. However, there are a few explanations for why the representation works so well. Here, following the notion of proposed concept-sample distribution (CSD), we propose a proposition to explain the training objective is to maintain the similarities relationship between samples in different modalities:

**Proposition 1** The pretraining task of VLMs **?**, contrastive image-text matching, is to minimize two concept-sample distributions with a shared concept $\mathbf{e}_i$ between different modalities sample set $\mathcal{I}$ and $\mathcal{T}$ given image-text pair $(x_i, t_i)$.

**Main steps of the proof.** The proof is done in three steps: In the first step, we formalize the training objective of contrastive image-text matching with the notion of cross entropy at the dataset level. In the second step, we dive into the sample-level perspective and transform the formulation into the subtraction of information entropy and KL divergence. Then we will find the minimized objective is just a KL divergence. In the third step, take the place of embedding of image and text in each pair with the shared concept of the pair

**Formalization.** Let $x \in \mathcal{X}$ denote an image $x$ in an image set $\mathcal{X}$, and $t \in \mathcal{T}$ represents a text $t$ in a text set $\mathcal{T}$. $\{(x_i, t_i) | i = 1, \ldots, N\}$ denotes $N$ image-text pairs, the match relationship can be represented with an identical matrix $\mathbf{Y}$, where $\mathbf{Y}_{ij} = \begin{cases} 1, & \text{if } i = j \\ 0, & \text{otherwise} \end{cases}$. In general, a VLM consists of an image encoder $I(\cdot)$, which maps the input image $x$ into a $d$-dimensional embedding space $\mathbb{R}^d$, and a text encoder $T(\cdot)$ which maps the input text $t$ into $\mathbb{R}^d$. We can get an image embedding matrix $\mathbf{I} = [I(x_1), \ldots, I(x_N)]$ and a text embedding matrix $\mathbf{T} = [T(t_1), \ldots, T(t_N)]$, where $\mathbf{I}, \mathbf{T} \in \mathbb{R}^{N \times d}$. The model is trained to maximize the similarity between the embeddings of matching image and text pairs.

$$\min_{I,T} \left[ H(\sigma(\frac{\mathbf{I} \cdot \mathbf{T}^\top}{\tau}), \mathbf{Y}) + H(\sigma(\frac{\mathbf{T} \cdot \mathbf{I}^\top}{\tau}), \mathbf{Y}) \right] \tag{1}$$

where $\sigma$ is the softmax operation applied in each row , $H(\cdot, \cdot)$ is the cross-entropy function $H(p, q) = -\sum_i p(i) \log q(i)$, and $\tau$ is a learnable temperature coefficient.

**Transform cross-entropy to KL divergence.** Consider each sample pair $i$, $\mathbf{Y}_i$ is actually an *one-hot* embedding, which can be viewed as parameters of a categorical distribution, where $\mathbf{Y}_{ij}$ is the probability of the $j$-th sample pair. Then the objective can be transformed as:

$$\min_{I,T}\left[H(\sigma(\frac{\mathbf{I}\cdot\mathbf{T}^\top}{\tau}),\mathbf{Y})+H(\sigma(\frac{\mathbf{T}\cdot\mathbf{I}^\top}{\tau}),\mathbf{Y})\right] \tag{2}$$

$$=\min_{I,T}\left[-\frac{1}{N}\sum_i^N\sum_j^N\mathbf{Y}_{ij}\left[\log\left(\sigma(\frac{\mathbf{I}\cdot\mathbf{T}^\top}{\tau})_{ij}\right)+\log\left(\sigma(\frac{\mathbf{T}\cdot\mathbf{I}^\top}{\tau})_{ij}\right)\right]\right] \tag{3}$$

$$=\min_{I,T}\left[-\frac{1}{N}\sum_i^N\sum_j^N\mathbf{Y}_{ij}\frac{\log\mathbf{Y}_{ij}}{\log\mathbf{Y}_{ij}}\left[\log\left(\sigma(\frac{\mathbf{I}\cdot\mathbf{T}^\top}{\tau})_{ij}\right)+\log\left(\sigma(\frac{\mathbf{T}\cdot\mathbf{I}^\top}{\tau})_{ij}\right)\right]\right] \tag{4}$$

$$=\min_{I,T}\left[-\frac{1}{N}\sum_i^N\sum_j^N\mathbf{Y}_{ij}\left[\log\left(\sigma(\frac{\mathbf{I}\cdot\mathbf{T}^\top}{\tau})_{ij}\right)+\log\left(\sigma(\frac{\mathbf{T}\cdot\mathbf{I}^\top}{\tau})_{ij}\right)\right]\right] \tag{5}$$

$$=\min_{I,T}\left[\frac{1}{N}\sum_i^N\sum_j^N\mathbf{Y}_{ij}\left[\log\frac{\mathbf{Y}_{ij}}{\sigma(\frac{\mathbf{I}\cdot\mathbf{T}^\top}{\tau})_{ij}}\right]+\frac{1}{N}\sum_i^N\sum_j^N\mathbf{Y}_{ij}\left[\log\frac{\mathbf{Y}_{ij}}{\sigma(\frac{\mathbf{T}\cdot\mathbf{I}^\top}{\tau})_{ij}}\right]-2\mathbf{H}(\mathbf{y})\right] \tag{6}$$

$$=\min_{I,T}\left[\frac{1}{N}\sum_i^N\mathrm{KL}(\mathbf{Y}_i\|\sigma(\frac{\mathbf{I}\cdot\mathbf{T}^\top}{\tau})_i)+\sum_j^N\mathrm{KL}(\mathbf{Y}_i\|\sigma(\frac{\mathbf{T}\cdot\mathbf{I}^\top}{\tau})_i)-2\mathbf{H}(\mathbf{y})\right] \tag{7}$$

In the above equation, $-2\mathbf{H}(\mathbf{Y}_i)$ is the information entropy, which is a constant for any $\mathbf{Y}_i$. We will omit it in the next step.

**Take place embedding with concepts** Intuitively, the image and text in an image-text pair express the same concept, which is the reason why CLIP uses language embedding as supervision. We denote the concept as a vector $\mathbf{e}_i=\mathbf{T}_i/\tau$, take the place of embedding (either image or text) as follows:

$$\sigma(\frac{\mathbf{T}\cdot\mathbf{I}^\top}{\tau})_i=\sigma(\mathbf{e}_i\cdot\mathbf{I}^\top).$$

Recall the definition of concept-sample distribution.

**Definition 2** (Concept-sample distribution). Given a sample set $\mathcal{Z}=\{\mathbf{z}_1,\mathbf{z}_2,\ldots,\mathbf{z}_n\}$ and a concept embedding $\mathbf{e}\in\mathbb{R}^d$, the concept-sample distribution (CSD) is defined as a categorical distribution over the sample set $\mathcal{Z}$ with following probability density function:

$$\delta(k;\mathbf{e},\mathcal{Z})=\frac{\exp(\mathbf{e}\cdot\mathbf{z}_k)}{\sum_{\mathbf{z}\in\mathcal{Z}}\exp(\mathbf{e}\cdot\mathbf{z})}=\sigma(\mathbf{e}\cdot f(\mathcal{Z})^\top)_i, \tag{8}$$

where $\mathcal{Z}$ can either be a text set or an image set. For convenience, we denoted CSD as $\delta(\mathbf{e},\mathcal{Z})$.

Formally, Eq 1 is equivalent to the following objective:

$$\min_{\mathbf{I},\mathbf{T}}\sum_i^N\left[\mathrm{KL}(\mathbf{Y}_i\|\delta(\mathbf{e}_i,\mathcal{T}))+\mathrm{KL}(\mathbf{Y}_i\|\delta(\mathbf{e}_i,\mathcal{X}))\right]. \tag{9}$$

Note that, we have assumed the $\mathbf{e}_i=\mathbf{T}_i/\tau=\mathbf{I}_i/\tau$ from the intuition of image-text matching to explain the training objective is to maintain the similarity across different modalities. The similarity relationship between image representations is trained to be similar to that between text representations so that the semantics of text can be adapted to images. However, the hypothesis is not guaranteed for unknown samples, which means $\|\mathbf{T}_i/\tau-\mathbf{I}_i/\tau\|>\epsilon$, leading to a shift between modality similarities. That is one of the reasons why we can not use concepts from text to build a VLM-based concept bottleneck models. Next, we will visualize the shift through CUB datasets.

# I  Concept-sample Distribution shift

As mentioned in the introduction of the main text, using textual concepts can introduce many biases due to the modality gap. In Figure 6, we visualize the distribution of features and concepts using t-SNE to demonstrate the biases that may exist in the concepts. We used 60 common image

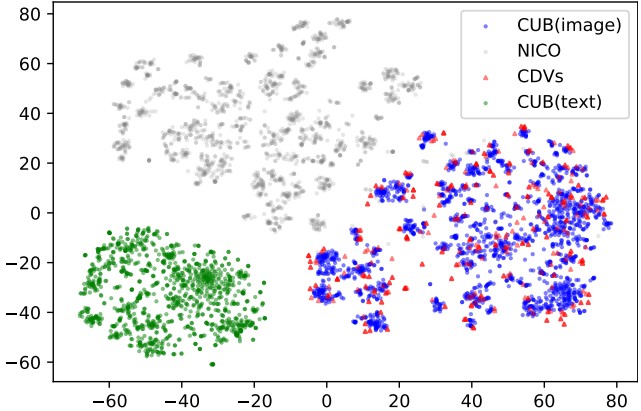

Figure 6: T-SNE visualization of latent features of CUB image, CUB related text, and CDV, with general image embeddings from NICOPP as background distribution.

categories(Including rabbits, birds and many other common object types) from the NICOPP dataset as the background distribution and then displayed the sample image feature distribution, textual concept feature distribution, and CDVs distribution for the CUB dataset. The results show that the distribution of textual features is completely different from that of image features, indicating a significant bias between textual concepts and image samples. In contrast, CDVs' distribution is similar to that of the image features, thus using CDV as concepts can effectively reduce biases in CBM and improve the model's classification performance.

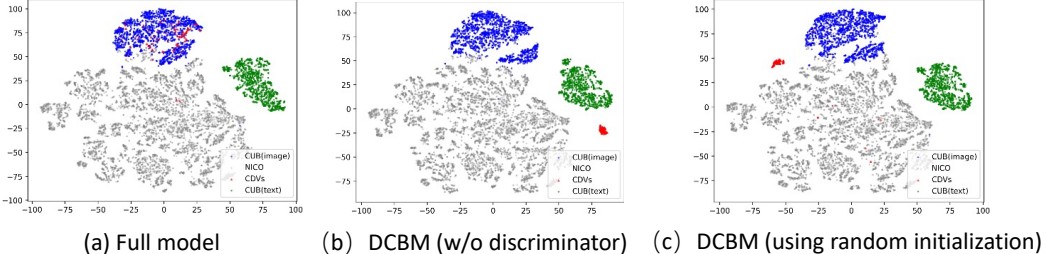

(a) Full model      (b) DCBM (w/o discriminator)    (c) DCBM (using random initialization)

Figure 7: Investigating the influence of discriminator $\Phi(\cdot)$ and initialization via t-SNE visualization. The text embedding of CUB-related corpus (the green dots), the image embedding of CUBs (the blue dots), and learned CDVs of CUB birds (the red dots) are shown simultaneously with image embedding from other 60 classes from NICO datasets as background (the gray dots). (a) In the full model, we observe that (1) image embeddings have a significant gap with texts. (2) the distribution of learned CDV highly overlaps within CUB image embeddings, indicating they are successfully constrained to express the visual semantics of birds. (b) without the discriminator, the learned CDV is far from both the image and text embeddings of CUB and lost their visual semantics. (c) using random initialization, there are only a few CDV aligned with image embeddings when the model has the best classification performance.

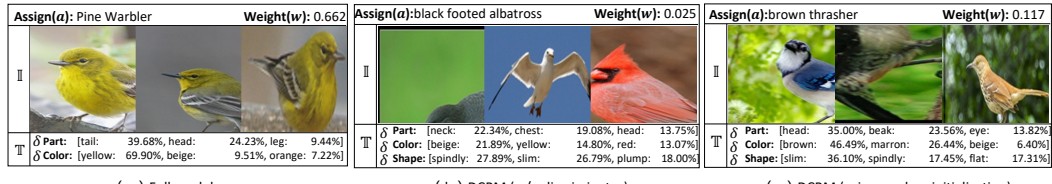

( a ) Full model      ( b ) DCBM (w/o discriminator)    ( c ) DCBM (using random initialization)

Figure 8: Influence of discriminator $\Phi(\cdot)$ and initialization via concept-level interpretation. We randomly selected the interpreted concepts. Compared to the full model (a), the CDV learned without $\Phi(\cdot)$ (b) shows inconsistent visualization results, and its interpreted probabilities of texts are nearly uniform, which indicates a meaningless embedding but accounts for better classification performance. (c) with the random initialization, the discriminator constrains the embedding, the visualization shows inconsistency while the text probabilities are slightly better than (b), which indicates the initialization strategy also has an impact on interpretability.