# OpenReview forum: "Cross-modality Interpretable image classification via Concept Decomposition Vector of Visual Language Models"
_ICLR.cc/2024/Conference — Submitted to ICLR 2024_

### Official Review · Reviewer_8g7G · 2023-10-29

**Soundness:** 2 fair
**Presentation:** 2 fair
**Contribution:** 2 fair
**Rating:** 6
**Confidence:** 4

**Summary:**

This paper addresses interpretability of image classifiers by leveraging vision-language models such as CLIP.  This paper modifies the explanations from Concept Bottleneck Models (CBM) by replacing pre-defined text from CBM with decomposed visual concepts that are not predefined, but instead learned. This method is denoted as Decomposed CBM (DBCM) in the paper.  The decomposed concepts are represented using vectors called CDV (concept decomposition vector). Experiments are conducted on a total of 9 datasets belonging to four types (natural images, semantic images, fine-grained images, and specialized application images).

**Strengths:**

1. Interpretability is an important problem in machine learning and this paper takes a novel approach in that direction, especially leveraging the recent advances in vision-language modeling.
2. Experiments are exhaustive and cover 9 datasets of 4 different types. Although results are not universally improved compared to prior work, having more evidence is useful.
3. The paper is presented well -- especially the figures 1,2 that provide the overview of the approach/method and help understand the differences with previous methods. The method is also sufficiently explained both in terms of intuition and mathematical/algorithmic formulation.
4. Sec 4.2 is very informative and the findings and analyses could be useful for future work.
5. In my opinion, Sec 4.3 is where this paper has the most potential for impact -- in terms of the 6 metrics, CDV clearly outperforms prior methods.

**Weaknesses:**

1. Having an algorithm pseudocode will further help the presentation of the paper.
2. In many cases, linear probe is the best performing method, but the highlighted numbers of Table 1 seem to ignore this. Why is that? I did not find any explanation for this.
3. The human study setting is lacking detailed description.  First, why is diverse geography important (or the right choice) given that the interpreted concepts are in English -- potentially English speaking countries (US, UK, Australia, India, Canada, etc.) could have been more appropriate?
4. n=27 sample size for the human study seems limited -- although the results from Fig 8 are interesting, I'm not sure if they are statistically significant.

**Questions:**

1. I did not find an explanation for the * after linear probe in Table 1.  What does the star * denote?
2. Could you show average performance across all nine datasets? By my rough calculations, linear probe seems to be better by 1% on average for ViT-B-16
3. Fig 8: why isn't there a radar chart for Linear Probe?

---

### Official Review · Reviewer_CHsn · 2023-11-02

**Soundness:** 3 good
**Presentation:** 2 fair
**Contribution:** 2 fair
**Rating:** 5
**Confidence:** 3

**Summary:**

This work proposes a method for "interpretable" image classification building on top of a work called Concept Bottleneck Models (CBMs). The main improvement proposed to CBMs is replacing a text embedding matrix used as the basis for claims of "interpretability" with a co-learned concept matrix. The concept matrix is composed of visual concept vectors corresponding to image-patch + text-phrases in a CLIP latent space (or at least are trained to be) that they hope distinguish classes from one another. To actually "interpret" these visual concept vectors (they call CDV), the work proposes either a text composition approach via LLMs or prompt-conditioned segmentation. They test on several small/medium scale datasets and compare against other CBM approaches through a small-scale user study.

**Strengths:**

I felt like the "tuning" of the "interpretable" embeddings that one uses in CBM closer to the instance/domain-under-test makes a lot of sense. This is both a strength and a weakness though, because it then puts the burden on the authors to prove that the previous "interpretable" embeddings truly were insufficient and would likely be insufficient in the future, and I feel like this actually wasn't shown/proven in the paper, e.g. is there a strong reason to believe that `T` in Eq. (3) won't just become good with better/larger VLMs a la Bitter Lesson [*].

[*] http://www.incompleteideas.net/IncIdeas/BitterLesson.html

**Weaknesses:**

* I generally felt the results were underwhelming. I'm glad a user study was done, but it's also hard to give it much weight without completely understanding the incentive structures as part of the survey, i.e. necessary but not sufficient. I wonder if the authors could've constructed a synthetic dataset (or found an existing one) that further enables validating the approach, and showcases where other methods fail.
* Looking at the text interpreted results across all examples, the text actually seems highly redundant, e.g. "brown" in Fig. 6 or "material of glass" in Fig. 5. Likewise, in the image segment interpretations, I couldn't tell qualitatively from the examples how they really distinguished from each other or would give confidence to a model developer, regulator, or user that the model was paying attention to the "concepts" that uniquely form the target class that aren't erroneous.
* The paper presentation was somewhat hard to follow in places (flow, grammar, etc). For example, Fig. 3 I gave up on trying to understand as it ended up confusing me more than just trying to understand the math formalization.

**Questions:**

It would help for the authors to briefly re-articulate the benefits of their method relative to other approaches *without* referencing the potential "bias" or "toxic" representations that may be in the text embedding, or else provide a more concrete case why one wouldn't expect these issues with the text embedding embedding wouldn't just get solved over time with a better model (indeed at the end of the paper, the authors acknowledge that this work was only validated with CLIP, so what if there was much much more powerful CLIP?). Answering this question would help clarify why the problem is important and hard.

---

### Official Review · Reviewer_VVq7 · 2023-11-04

**Soundness:** 2 fair
**Presentation:** 2 fair
**Contribution:** 3 good
**Rating:** 5
**Confidence:** 4

**Summary:**

This paper presents an alternative approach to interpretable image classification through the introduction of the Decomposed Concept Bottleneck Model (DCBM), a model designed to circumvent the limitations of traditional text-based interpretative methods. Due to the reliance on pre-defined textual concepts, existing works show compromised interpretability and performance. DCBM seeks to rectify this by learning visual concepts directly from images and employing these for classification tasks. Concept decomposition is executed by projecting image features onto Concept Decomposition Vectors (CDVs), with the aim of distilling critical visual elements integral to the classification. To articulate these concepts across modalities, the paper introduces a quintuple notion of concepts along with a concept-sample distribution technique, positing that this method enhances interpretability in various dimensions such as sparsity, groundability, factuality, fidelity, and meaningfulness.

**Strengths:**

Clarity of Conceptual Innovation: The idea of the Decomposed Concept Bottleneck Model (DCBM) is articulated with notable clarity, making a case for the paradigm shift from text-based to visual concept interpretations in image classification. The DCBM is well-conceived, addressing a recognized gap in interpretable machine learning by leveraging visual concepts that are intrinsically more aligned with the modality of the data being processed. This approach has the potential to significantly improve the interpretability of image classification systems by utilizing concepts that naturally reside within the visual domain, thus allowing for a more intuitive understanding of the model's decision-making process.

Commitment to Empirical Validation: The paper showcases a commendable breadth of experimental work aimed at validating the proposed DCBM framework. The extensive experiments are designed to evaluate the model's performance and interpretability, endeavoring to benchmark against non-interpretable models and other CBMs. Through this exhaustive testing, the research demonstrates a commitment to not just proposing a theoretical model but also to empirically substantiating its effectiveness across several metrics. This extensive experimental evaluation provides a foundation for understanding the practical implications of the proposed method, suggesting its potential to compete with, and possibly exceed, the current state-of-the-art in certain interpretability aspects.

**Weaknesses:**

1. Although Figure 2 shows some motivating failure examples for text concepts, some of them seem to be solved by prompting engineering, e.g., "what are useful visual features for distinguishing a blue Grosebeak in a photo", "eliminating the answers with the concept cannot observe from an image".
2. The performance improvement is relatively minor as compared to Labo. Although the interpretability is the key idea, while the user study with 27 users shows significant improvement, the results may be very subjective and sensitive to the selected samples. It is suggested to propose some new metrics that can automatically measure the interpretability.
3. Important references are missing, e.g., [A] and its subsequent works. It is suggested to compare with it empirically and theoretically.
4. The references are out-of-date, i.e., only two papers published in 2023 are cited.

[A] Visual Classification via Description from Large Language Models. ICLR, 2023.

**Questions:**

It is suggested to proofread the paper carefully for a better readability. To name a few, there is an unfinished sentence in Introduction, i.e., “The adversarial training ensure CDVs”. Moreover, ϵ is undefined in the method and unspecified in the experiments. Eq should be at least corrected as "Eq.". Finally, the references are incomplete, even for important references. For example, (Oikarinen & Nguyen, 2023) is without the publication venue (ICLR 2023) and with an incomplete author list (only two authors).
https://openreview.net/forum?id=FlCg47MNvBA

---

### Official Review · Reviewer_U8yY · 2023-11-09

**Soundness:** 1 poor
**Presentation:** 1 poor
**Contribution:** 2 fair
**Rating:** 1
**Confidence:** 3

**Summary:**

The paper introduces Concept Decomposition Vectors (CDVs), which are essentially vector embeddings of a 'concept' in a visual language model such as OpenAI's CLIP. These CDVs are learned as part of a concept bottleneck model, which they call Decomposed Concept Bottleneck Model. The CDVs, being embedding vectors, are then matched to corresponding images and text, which supposedly makes the CDVs explainable.

**Strengths:**

The paper presents an attempt to create 'concepts' that are not manually defined by humans, which many other studies rely on. This may be promising if the approach is improved upon.

**Weaknesses:**

I had a difficult time understanding the paper. The paper is poorly written and needs a major overhaul. Apart from being replete with grammatical errors, misspellings and confusing vocabulary, the paper is incomplete. This is most evident in Section 4.4, where the experiments that were supposedly performed aren't described and the paragraph refers to a missing appendix.

**Questions:**

Can the authors address the following:
1. What regularization loss did you use in your experiments?
2. Equation 7: What is epsilon? It is not defined anywhere.
3. What is a concept-sample distribution? What is the intuition?
4. Section 4.3: Can you describe what the objective of the experiment and how is the experiment set up? Calling it an 'intuitive comparison' between models helps no one. In addition, can you describe how you collected the human labels? It would be good to show an example of the survey, describe the filtering criteria of the candidates (if any), detail the questions asked, and show the distribution of the answers and the associated uncertainties.

I would suggest that the authors rework the paper and have it proofread before resubmitting.

---

### Public Comment · ~Zhengqing_Fang1 · 2024-05-21
**A revised version has been published!**

A revised version has been published in the IEEE Transactions on Circuits and Systems for Video Technology (TCSVT).

For reference, see the document at IEEE Xplore.
https://ieeexplore.ieee.org/document/10535313

---

### Meta-Review · Area_Chair_WiGa · 2023-12-09

**Metareview:**

This paper propose decomposed concept bottleneck model, an alternative approach leverage visual concept vectors instead of predefined texts. Despite the idea is interesting, the reviewers generally found that the quality of paper is not high (e.g. poorly written / low readibility, unfinished sentences, incomplete and outdated reference, concerns in experiment setup and results). As the authors did not submit rebuttal in response to the reviewers' concerns, rejection is recommended.

**Justification For Why Not Higher Score:**

paper is of low quality

**Justification For Why Not Lower Score:**

N/A

---

### Decision · Program_Chairs · 2024-01-16

Reject